# Nephroprotection by SGLT2 Inhibition: Back to the Future?

**DOI:** 10.3390/jcm9072243

**Published:** 2020-07-15

**Authors:** Luca De Nicola, Francis B. Gabbai, Carlo Garofalo, Giuseppe Conte, Roberto Minutolo

**Affiliations:** 1Nephrology Division, Department of Advanced Medical and Surgical Sciences, University of Campania Luigi Vanvitelli, 80138 Naples, Italy; carlo.garofalo@unicampania.it (C.G.); giuseppe.conte@unicampania.it (G.C.); roberto.minutolo@unicampania.it (R.M.); 2Department of Medicine, VA San Diego Healthcare System, University of California at San Diego Medical School, San Diego 92103, CA, USA; fgabbai@ucsd.edu

**Keywords:** SGLT2-inhibition, chronic kidney disease, nephroprotection

## Abstract

The introduction of sodium/glucose cotransporter 2 inhibitors (SGLT2i) has opened new perspectives for the management of diabetic population at risk of or with chronic kidney disease (CKD). More important, recent, large real-world studies have repositioned the nephroprotective efficacy of SGLT2i emerged from randomized trials within the frame of effectiveness. Furthermore, the salutary effects of these agents may extend to the nondiabetic population according to the positive results of current studies. Nevertheless, the clear benefits of these agents on the prevention of organ damage contrast with their unexpected, limited use in clinical practice. One potential barrier is the acute decline in glomerular filtration rate (GFR) commonly observed at the beginning of treatment. This phenomenon is reminiscent of the early response to the traditional nephroprotective interventions, namely blood pressure lowering, dietary protein and salt restriction and the inhibition of the renin–angiotensin system. Under this perspective, the “check-mark” sign observed in the GFR trajectory over the first weeks of SGT2i therapy should renew interest on the very basic goal of CKD treatment, i.e., alleviate hyperfiltration in viable nephrons in order to prolong their function.

## 1. Introduction

In the last decade, the epidemiology of chronic kidney disease (CKD) and focus on the nephrology community have changed in parallel. Two major changes have occurred. First, the CKD population has grown, due not only to the expression of the demographic expansion in developing countries, but also to the ageing of the general population in developed countries, with CKD associated with increased incidence of geriatric diseases (hypertension, heart failure and diabetes) [1]. Second, referral to renal clinics has improved due to increased awareness of the heavy burden of CKD and the mounting evidence regarding the efficacy of nephrology care among patients and non-nephrology providers [2,3,4]. In contrast, a high mortality rate and poor quality of life on dialysis remain unresolved issues despite improved technology [5]. Hence, a need to rethink the approach to CKD from replacing to preserving renal function has emerged [6]. In this context, the development of the inhibitors of sodium/glucose cotransporters 2 (SGLT2i) represents a major step forward in the research of novel nephroprotective agents for diabetic CKD, and, possibly, for non-diabetic CKD.

The purpose of this review is twofold: (1) analyze the information supporting the hypothesis that the clinical effectiveness of SGLT2i is mainly related to the attenuation of hyperfiltration; and (2) carefully review data on SGLT2i nephroprotection to highlight similarities between SGLT2i and old strategies to reduce renal hyperfiltration (Renin Angiotensin System -RAS- inhibition, strict blood pressure -BP-), control, low-protein–low-salt diet).

## 2. The New Era of Nephroprotection in Diabetic Patients: From Efficacy to Effectiveness of SGLT2i

A recent review by Chan and Tang has clearly described two main unmet needs in nephrology practice [7]: (1) the limited efficacy and/or low tolerability of traditional nephroprotective interventions, as demonstrated in as many as 23 landmark randomized trials (RCT) in diabetic patients with and without CKD; (2) the absence of clinical benefits in the patients of new agents with nephroprotective efficacy proven in experimental models. Therefore, therapy with RAS inhibitors (RASi) remains the basic nephroprotective intervention which, unfortunately, still leaves 30% to 50% of patients with a high risk of end-stage kidney disease (ESKD) [7]. Accordingly, the introduction of SGLT2i appears to be a major step to reduce the burden of ESKD. The beneficial renal effects of SGLT2i were first shown in the cardiovascular outcome trials (CVOTs) EMPAREG, CANVAS and DECLARE. These trials were originally aimed at evaluating cardiovascular safety in patients with type 2 diabetes (DM2), with renal outcomes as secondary endpoint [8,9,10]. The large majority of patients were at low risk of kidney failure and few renal events were observed. A major step forward was the CREDENCE trial, specifically aimed, and therefore powered, to assess renal survival in a large population with DM2 with overt CKD [11].

Overall, the results of the four large RCTs demonstrated, along with cardiovascular benefits [12,13], a major nephroprotective efficacy of SGLT2i, with a significant 30% reduction in albuminuria and a 30–40% lower risk of progression to ESKD. These beneficial effects were further enhanced by optimal RAS inhibition as background therapy in most patients.

Of note, SGLT2i-nephroprotection was not encumbered by significant adverse effects. In particular, the incidence of acute kidney injury (AKI) and hyperkalemia was not higher in the four CVOTs on SGLT2i [8,9,10,11], thus emphasizing the added value of a safe renal profile of these new agents. This aspect is important when considering that AKI and hyperkalemia are the main determinants of withdrawal of RASi treatment, with the consequent potential worsening of kidney disease and increased cardiovascular risk [14,15].

On the other hand, when compared to other antihyperglycemic agents, SGLT2i are associated with a 3-fold increase in risk of genital infections [16], and a small, but consistently greater, incidence (two to five events per 1000 person-years) of euglycemic diabetic ketoacidosis (euDKA) [17]. Appropriate information of these side effects should be provided to the patient at the time of prescription to prevent these side effects from becoming a major barrier to the proper use of these agents in eligible patients (education on appropriate personal hygiene and adequate daily water intake to avoid infections and drug withdrawal during intercurrent illness or surgery -the so called “sick day”- to avoid euDKA).

It is noteworthy that evaluating effectiveness above and beyond efficacy is needed when examining a new drug. In this regard, the experience with spironolactone in heart failure (HF) provides a great example. The Randomized Aldactone Evaluation Study (RALES) study demonstrated that spironolactone significantly reduced the risks of death and hospitalization in HF patients [18]. In the years following the study, the use of this agent by practicing cardiologists resulted in a peak in hyperkalemia-associated morbidity and mortality in the HF population [19]. In contrast, there is no discrepancy between efficacy and effectiveness with SGLT2i. Indeed, the Comparative Effectiveness of Cardiovascular Outcomes in New Users of SGLT-2 Inhibitors (CVD-REAL) studies using data from claims databases, medical records, and national registries from various countries have proven the effectiveness of SGLT2i compared to other glucose-lowering agents in large populations. The initial CVD-real data confirmed superiority of SGLT2i for cardiovascular risk across a broad range of outcomes and patient characteristics [20,21,22], with the beneficial effects being even larger compared to the original CVOTs. More recently, the first real-world data on the effects of SGLT2i on renal outcomes were provided by the CVD-REAL 3 [23]. More than 35,000 patients initiating SGLT2i were matched to a similar number of patients starting other glucose-lowering drugs. During 15 months of follow-up, SGLT2i treated patients had slower eGFR decline vs. controls (difference in slope 1.53 mL/min/1.73 m² per year, 95% CI 1.34–1.72) and lower incidence of the composite outcome of eGFR decline ≥50% or ESKD (hazard ratio 0.49, 95% CI 0.35–0.67). Interestingly, in CVD-REAL 3 mean eGFR was 91 mL/min/1.73 m² at baseline with as many as 92% patients with eGFR >60 mL/min/1.73 m², thus demonstrating that management of these patients could be considered as the primary prevention of kidney disease. Similar findings have been recently reported by Pasternalk and colleagues using pooled data from Scandinavian registries [24]. Using propensity score analysis, as in CVD-REAL, they matched new users of SGLT2i (*n* = 29,887) with patients receiving dipeptidyl peptidase 4 inhibitors (DPP4i) (*n* = 29,887). As in CVD-REAL3, only a minority (3%) had CKD at baseline. Analysis showed that SGLT2i were associated with 58% (95% CI 47% to 66%) lower risk of the composite renal endpoint (renal replacement therapy, hospital admission for renal events, or death from renal causes) compared to DPP4i.

The information emerging from real-life practice is as important as that generated from randomized trials, since individuals selected for SGLT2i trials can be poor representatives of the “generic” patients seen in daily practice [25]. Therefore, these real-word studies have allowed for repositioning the results on an SGLT2i-related nephroprotection within the frame of effectiveness.

## 3. Nephroprotection by SGLT2i in Diabetic Patients: From Bench to Bedside

Understanding the mechanism(s) underlying the beneficial effects of SGLT2i on renal survival is crucial to enhance the confidence of physicians toward these new drugs. Indeed, despite the solid evidence of cardio- and nephro-protection, the current prescription is still modest in daily practice (<10% of eligible patients) even six years after their initial marketing [20,21,22,23,24].

The nature of diabetic kidney injury is complex with the involvement of hemodynamic and nonhemodynamic factors primarily activated by hyperglycemic milieu [26]. Based on the multifactorial pathophysiology of diabetic kidney disease (DKD) and the striking SGLT2i-related nephroprotection, several mechanisms have been proposed to explain the remarkable renal benefits of this new class of agents. Systemic effects to consider include the reduction in extracellular volume (ECV), total body sodium content, and arterial stiffness leading to lower blood pressure (BP) and albuminuria [26,27,28,29]. Furthermore, besides the antihyperglycemic effect and associated reduction in glucotoxicity, SGLT2i may improve endothelial function via several mechanisms including weight loss and decreased body fat due to daily energy losses of up to 300 kcal (related to glycosuria 70–80 g/day), decreased insulin resistance and reduced uric acid levels [26,27,28,29]. More recent data suggest a role for the reduction in oxidative and endoplasmic reticulum stress due to the increment in autophagic flux in podocytes and renal tubules [30]. Of relevance are the anti-fibrotic or anti-inflammatory effects of SGLT2i. Indeed, studies in two independent human proximal tubular cell lines have recently demonstrated that SGLT2i block basal and TGF-β1-induced expression of key mediators of renal fibrosis and kidney disease progression, namely thrombospondin 1, tenascin C and platelet-derived growth factor subunit B [31]. Interestingly, these experimental results were obtained under normoglycemic conditions, suggesting that the SGLT2i-induced antifibrotic effects at the cellular level are independent from diabetic status.

Noteworthy, the reduction in glucose reabsorption with SGLT2i is associated with significant changes in renal hemodynamics. Micropuncture studies performed in hyperglycemic diabetic rats demonstrated that poor glucose control is associated with increased GFR (hyperfiltration) at the whole kidney and single nephron level. The presence of hyperfiltration is now recognized as a major mechanism of diabetes-induced renal injury in both humans and experimental animals [32]. Based on kidney micropuncture studies, Vallon, Thomson and Blantz have proposed the “tubulocentric” hypothesis to explain the hemodynamic responses of the kidney to an increased glucose load as well as the beneficial effects of SGLT2i [33]. Briefly, diabetes promotes proximal tubular cells hypertrophy with a consequent increased expression of SGLT2 leading to increased proximal tubular reabsorption and decreased distal delivery of sodium chloride to the macula densa. Decreased distal delivery deactivates the tubuloglomerular feedback (TGF) system responsible for modulating nephron filtration depending on the amount of sodium chloride reaching the macula densa [34]. Indeed, decreased distal delivery causes dilation of the glomerular afferent arteriole, which allows nephron filtration to increase. Recent experimental data have shown that the delivery of glucose to macula densa also activates SGLT receptors located in this structure, with the consequent stimulation of nitric oxide generation via the neuronal nitric oxide synthase [35]. The generation of such potent vasodilator promotes afferent dilation and increases in nephron filtration, providing an additional mechanism underlying the modulation of afferent glomerular resistance in conditions of poorly controlled glycemia.

SGLT2i restore proximal tubule flow rate by counteracting “hyper-reabsorption”. The reduction in proximal sodium reabsorption increases delivery to macula densa, thus reactivating TGF and restoring a normal (lower) nephron filtration and, likely, intra-glomerular capillary pressure. In addition, SGLT2i also inhibits sodium/hydrogen exchanger 3 in the proximal tubule, another potential contributor to higher distal delivery of sodium chloride [36]. Finally, it is also possible that the mild ECV depletion induced by SGLT2i further attenuates hyperfiltration by making the TGF system more sensitive to sodium chloride [34].

The TGF-dependent modulation of afferent arteriolar tone, with and without SGLT2i, has been confirmed in vivo by the direct visualization of arterioles in diabetic mice [37]. Some supporting evidence has also been obtained in patients. Indeed, a study published by Parving group in 1988 demonstrated that 3-day treatment with acetazolamide (i.e., a diuretic that, similarly to SGLT2i, acts at the proximal tubular level) in type 1 diabetics produced a decline in fractional proximal reabsorption of water and sodium, 10% reduction of extracellular volume (ECV) and, more importantly, a 24% GFR reduction associated with a significant lowering of albuminuria [38].

Therefore, SGLT2i alleviates glomerular hypertension and hyperfiltration, thus reducing glomerular barotrauma and albuminuria. Of note, a potential additional consequence of attenuation of hyperfiltration is the decrease in oxygen (O2) demand for the active tubular transport of solutes, sodium in primis, with improved renal tissue oxygenation and prevention of fibrosis [39,40].

## 4. SGLT2i Nephroprotection Independent of Diabetic Status

SGLT2i-related cardionephroprotection is, for the most part, independent from the glucose lowering effect. This was consistently reported in the first three CVOTs enrolling patients with mild or no kidney disease [8,9,10] and confirmed by the CREDENCE trial in overt DKD [11]. Overall, these trials demonstrate that, in the more advanced stages of CKD, the glucose-lowering capacity virtually disappears, due to the reduced filtered load of glucose, while the renal and cardiovascular benefits remain unaltered. Furthermore, a secondary analysis of CREDENCE demonstrated that canagliflozin reduced cardiorenal risk across the whole spectrum of baseline HbA1c values, including the ideal range of 6.5% to 7% [41].

In accordance with the dissociation between antihyperglycemic effect and organ protection, the recent Dapagliflozin and Prevention of Adverse Outcomes in Heart Failure (DAPA-HF) trial provides evidence that the salutary effects of SGLT2i can be extended to nondiabetic patients [42]. The reduction in the risk of the composite endpoint (worsening HF or cardiovascular death) in DAPA-HF was almost identical among the enrolled 2139 diabetic and 2605 nondiabetic patients (−25% and −27%, respectively). Furthermore, no interaction was found between HbA1c and outcome (*p* = 0.97). As additional proof of concept, the reduction in primary outcome with dapagliflozin among nondiabetic individuals did not differ between those with prediabetes and those with normal HbA1c. Whether a similar beneficial response among diabetic and nondiabetic patients also applies to kidney survival could not be adequately tested in this trial aimed at evaluating HF outcome. Nevertheless, the expectations regarding SGLT2i-nephroprotection in nondiabetic CKD are now becoming more realistic based on the recent report of early stop for “overwhelming efficacy” of the DAPA-CKD study [NCT03036150], that is, the phase III trial specifically designed to test the benefit of dapagliflozin on kidney survival in CKD patients independent of diabetic status [43]. Accordingly, SGLT2i seem to have crossed, almost by chance, the (already) crowded highway of glucose-lowering drugs, while properly entering the (still) empty and winding road of nephroprotective agents.

## 5. The “Check-Mark Sign” (√) in Patients Starting SGLT2i Therapy

Results of DAPA-HF and the early stop of DAPA-CKD suggest the interesting possibility of a unifying pathophysiological pathway for SGLT2i nephroprotection in diabetic and nondiabetic CKD. Noteworthy, in the dapagliflozin arm of diabetic and nondiabetic individuals enrolled in DAPA-HF, eGFR showed a biphasic pattern of changes, with an initial and transient dip of about 4 mL/min/1.73 m^2^ over the first two weeks of dapagliflozin followed by recovery to baseline and stabilization during the subsequent months. In contrast, eGFR progressively declined in control patients throughout follow-up [42].

The biphasic trajectory of eGFR of dapaglifozin-treated patients, either diabetic or nondiabetic, resembles a “check-mark” sign (√) because of the initial dip in eGFR followed by a raise of the levels and subsequent eGFR stabilization. This check-mark effect suggests reduced hyperfiltration at the beginning of SGLT2i treatment that translates into the preservation of kidney structure over the long term. This peculiar hemodynamic sign heralding better kidney outcome has been a consistent finding in all the early SGLT2i trials [8,9,10,11], as well as in the most recent VERTIS trial in diabetic patients treated with ertuglifozin [44].

The concept that check-mark sign is a class-effect of SGLT2i is supported by the similar biphasic eGFR trajectory observed in the broad population of CVD-REAL 3 patients treated in clinical practice with six different SGLT2 inhibitors [23].

The peculiar pattern of eGFR response to SGLT2i is consistent not only among the various agents used, but also in the presence and absence of DM2 and independent of the severity of kidney disease. This pattern is, in fact, present both in high eGFR strata and in patients with low eGFR. It was observed in the EMPAREG population [8], as well as in CREDENCE where it was associated with better renal outcome in patients with eGFR as low as 30–45 mL/min/1.73 m^2^ [45]. A similar association has been observed across different levels of baseline albuminuria [46], and in patients with and without background RASi therapy [47].

In contrast, no checkmark phenomenon becomes apparent when one examines the effects of glucagon-like peptide-1 receptor agonists (GLP1-RA). This discrepancy is relevant because the current guidelines recommend GLP1-RA as a therapeutic alternative to SGLT2i for organ protection in DM2 [48,49,50]. Indeed, trials have demonstrated that therapy with GLP-1RA reduces the risk of the composite renal endpoint (albuminuria, doubling of serum creatinine or decline in eGFR, end-stage kidney disease or death from a renal cause) by about 5–10%; however, this result is exclusively dependent on the antialbuminuric effect [12]. It is, therefore, possible that GLP-1RA may be nephroprotective by means of antialbuminuric effects other than changes in intrarenal hemodynamics, namely the reduction in glycemia, body weight, BP, endothelial dysfunction and inflammation. Accordingly, studies with a more prolonged follow up are needed to test whether the antialbuminuric effect of GLP-1RA translates into the long-term preservation of renal function.

## 6. The Check-Mark Sign: Not Only SGLT2i

Table 1 summarizes the findings of RCTs with available results on GFR changes, comparing early (first few months) with subsequent months of treatment in experimental and placebo arms. As previously mentioned, all major trials with SGLT2i demonstrate a checkmark effect that is consistently associated with better renal outcome [8,9,10,11,44]. Figure 1 is a graphical description of the checkmark phenomenon, reporting the mean eGFR changes within the initial six months of trials and in the subsequent period of follow-up in patients administered with an experimental drug or placebo.

Interestingly, the biphasic pattern of GFR changes is not an exclusive feature of SGLT2i since it can be observed, although with more heterogeneous results, in CKD patients undergoing nephroprotective interventions different from SGLT2i [51,52,53,54,55,56,57] (Table 1 and Figure 1). Specifically, the association between an initial transient dip in GFR and its preservation over time is a typical finding of CKD patients starting antihypertensive therapy. In 1992, a landmark randomized trial in type 1 diabetic patients with advanced CKD (basal GFR 47 mL/min and proteinuria 2.0 g/24 h) compared enalapril and metoprolol. Enalapril induced a greater GFR fall in the first six months, but a slower GFR decline and greater reduction in proteinuria over the subsequent 30 months (in the presence of similar BP control) [51]. A similar finding was later confirmed by a post-hoc analysis of the large RENAAL study [52]. The same result was also observed in the microalbuminuric subgroup of American Indians with DM2 where losartan led to an early decrease in GFR followed by a slower decline when compared to placebo [53]. The overall effect on whole GFR decline was not statistically significant, possibly due to the small sample size; however, RASi allowed for a greater reduction in mesangial fractional volume (*p* = 0.02). With regards to nondiabetic CKD, an observational study from the late 1990s in patients with a mean GFR 56 mL/min at baseline showed that the GFR fall in the first three months after starting either enalapril or atenolol was associated with lesser losses of GFR during the following four years [58]. This observation was further highlighted by Maschio et al. in the benazepril trial in CKD patients with average creatinine clearance 43 mL/min, where an early increase in serum creatinine after RAS inhibition heralded better preservation of renal function at the end of the trial [59]. Unfortunately, this trial did not report GFR values.

Noteworthy, the biphasic GFR trajectory is not limited to antihypertensive therapy. Indeed, the MDRD study A (GFR 25–55 mL/min) showed that the initial GFR decline after starting intervention (04–months) was significantly faster in the active arms (low-protein diet or low BP) when compared to controls [54]. After 4 months, GFR decline was 28% slower (*p* = 0.009) in the low-protein diet and 29% slower (*p* = 0.006) in the low-BP patients. These two changes in the opposite direction led to a non-significant difference in GFR when the entire follow-up was considered. It is possible that a longer follow-up would have been required to reach statistical significance. Interestingly, with regards to the acute GFR response to a low-protein diet, we have previously demonstrated by means of inulin and PAH clearance studies that, in CKD patients (basal GFR around 40 mL/min), reduced protein intake per se does not translate in acute decline in GFR if salt intake is kept constant [60]. In contrast, low salt intake in the presence of constant protein intake is associated with a GFR reduction of 6 mL/min on average related to reductions in ECV and BP [61]. These analyses, therefore, allowed us to hypothesize that the acute reduction in GFR observed after starting a low-protein diet is secondary to the (frequently) associated dietary salt restriction. Similarly, the acute decrease in GFR measured after bariatric surgery in obese individuals has been found to be dependent on the reduction in sodium rather than protein intake [62].

Of interest, other trials analyzing changes in GFR over time support the concept that the absence of GFR dip upon initiating antihypertensive treatment does not lead to long-term benefit in renal function [55]. Absence of the GFR dip can even be coupled with significant worsening in kidney function [56]. In this scenario, the results of the SPRINT-CKD appear unique as they show that the early decrease in GFR in the intensive BP arm heralded the worsening of GFR with a final, non-significant change in kidney survival [57]. These results, however, may have been influenced by the 46% increase in the risk of AKI. Indeed, this trial was not designed to evaluate renal survival in CKD and de facto enrolled patients from the general population where a low eGFR is probably reflected kidney senescence rather than primary kidney disease [63].

## 7. The Checkmark Sign: Pathophysiological Insights

As evident from Table 1 and Figure 1, the initial transient dip in GFR after starting a nephroprotective therapy is neither limited to SGLT2i nor to diabetes. The association between the “check-mark” and improved renal outcome is however more evident for SGLT2i. Whether this is merely dependent upon the larger number of patients included in the SGLT2i trials or due to the specific properties of this new class of agents, including a better safety profile, is unknown at this time. With regard to the latter hypothesis, it is interesting that the early dip is, on average, more pronounced after interventions other than SGLT2i (Figure 1), thus suggesting the possibility of greater risk of AKI from traditional therapy (antihypertensive therapy, low salt diet and RASi). Ad hoc studies should explore the hypothesis that the initial mild-to-moderate eGFR decline after starting SGLT2i allows for a more preserved renal function and structure in the long term.

Nonetheless, sodium, more than glucose, appears to be the key player in the SGLT2i-related nephroprotection and, accordingly, it is not surprising that the beneficial effects of these drugs are independent of the presence of diabetes mellitus. To better understand the nephroprotection coupled with this biphasic pattern of GFR changes, we need to remember that seminal studies in the nephrology field clearly demonstrated that, independently of the primary renal disease, loss of functional renal mass causes functional and structural hypertrophy in remnant nephrons [64,65]. This abnormality is a “*two-faced Janus*” allowing, on the one hand, the maintenance of homeostasis in the presence of reduced renal mass while, on the other hand, increasing the risk of progressive damage in surviving nephrons. Interventions aimed at attenuating hyperfiltration in the diabetic or diseased kidney may consequently turn into less progressive renal damage. Under this view, we may consider SGLT2i as a step back to the future.

## 8. Conclusions

In diabetes, tubular hypertrophy aimed at limiting losses of large amount of glucose, salt and water, leads to the development of intrarenal hemodynamic changes, such as glomerular hyperfiltration and hypertension, that are associated with the worsening of renal prognosis. SGLT2i counteract these changes by restoring a normal TGF activity and GFR, thereby reducing injury to glomeruli and tubulointerstitium (Figure 2).

The long-term favorable renal outcome is anticipated, not only by the reduction in albuminuria, but also by the acute transient GFR dip. In this scenario, the reduced proximal reabsorption of sodium may be a critical element to correct hyperfiltration both in diabetes and non-diabetic CKD. Past knowledge of the role of hyperfiltration in progression of diabetic and nondiabetic CKD may help to better understand SGLT2i nephroprotection. At the same time, the proven efficacy of these new agents may renew interest in the traditional approaches aimed at lowering glomerular hypertension and hyperfiltration. Further studies on old and new nephroprotective interventions are definitely needed to explore the association between early dip of GFR, as a proxy of attenuation of hyperfiltration in remnant nephrons, and subsequent risk reduction of hard renal endpoints.

The upcoming results of DAPA-CKD trial [NCT03036150] and the still recruiting EMPA-KIDNEY trial [NCT03594110] will hopefully provide conclusive information on the appropriate role of SGLT2i in the therapeutic algorithm aimed at optimizing nephroprotection in the whole CKD population, irrespective of diabetic status. Given the well-recognized pathophysiological role on CKD progression of hyperfiltration and albuminuria, it is plausible that in the future SGLT2i will be considered as the “ideal companion” of RASi, or, alternatively, replace these traditional nephroprotective agents in patients that do not tolerate them because of AKI and/or hyperkalemia.

## Figures and Tables

**Figure 1 jcm-09-02243-f001:**
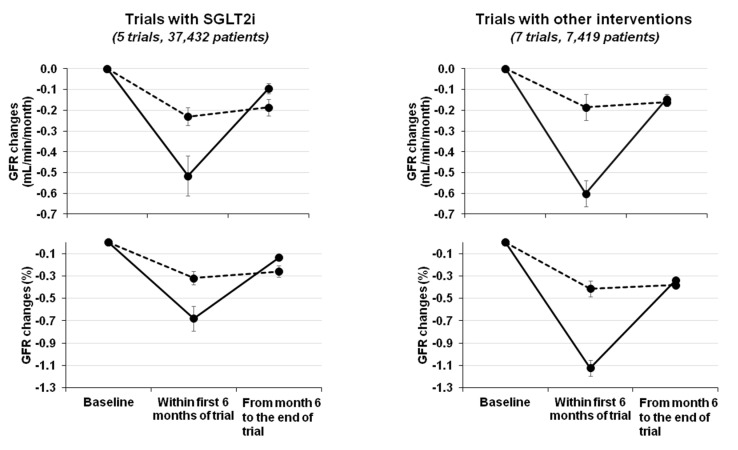
Check-mark sign in randomized trials reporting GFR values by period of follow up. Solid line represents changes in active arm, while dotted line refers to placebo arm. Trials are described in Table 1. GFR decline is absolute (upper panel) or percentual (lower panel) and calculated from baseline to the first 6 months and from 6 months to the end of trial. Absolute changes are calculated as weighted means in order to take into account the different sample size in trials, and are expressed as mL/min/month to take into account the different timing of GFR assessment in different trials. The initial period occurred after 5.1 ± 1.5 months and end of trial after further 43.3 ± 17.8 months for trials with SGLT2i and after 5.6 ± 3.1 months and end of trial after further 41.6 ± 13.8 months for trials with other interventions.

**Figure 2 jcm-09-02243-f002:**
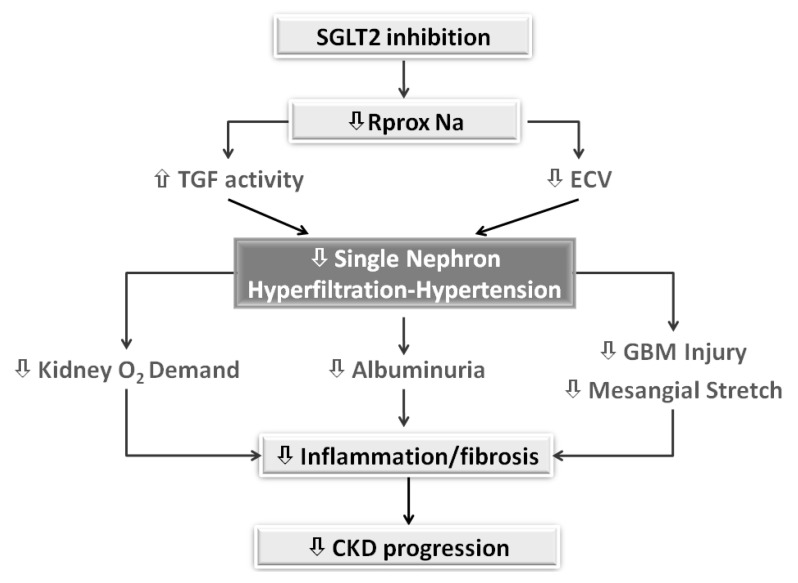
Intrarenal hemodynamic mechanisms of nephroprotection by SGLT2 inhibition. SGLT2, sodium/glucose cotransporter 2; Rprox Na, proximal tubular reabsorption of sodium; TGF, tubuloglomerular feedback; ECV, extracellular volume; GBM, glomerular basal membrane.

**Table 1 jcm-09-02243-t001:** GFR changes by period of follow up in the main randomized trials (RCTs) on SGLT2 inhibition in diabetic patients and in RCTs testing other interventions in chronic kidney disease (CKD) patients.

Trial ref	Active Arm	Control Arm	N (Active/Control) †	DKD	GFR/Scr	Time (Years)	eGFR Changes (mL/min/month)				P for Overall eGFR Decline °
							Active Arm		Control Arm		
							Within 6 Months	6 Months-End Trial	Within 6 Months	6 Months-End Trial	
SGLT2 Inhibition											
EMPAREG ^, [8]	Empaglifozin	Placebo	2322/2323	Yes	30–59	4.0	−0.21 *	−0.04	−0.03	−0.15	<0.001
CANVAS [9]	Canaglifozin	Placebo	5711/4276	Yes	>30	4.0	−0.50 *	−0.04	−0.18	−0.08	<0.001
DECLARE [10]	Dapaglifozin	Placebo	8581/8578	Yes	85 ± 16	4.0	−0.55 *	−0.13	−0.22	−0.21	<0.001
CREDENCE [11]	Canaglifozin	Placebo	2179/2178	Yes	30–90	3.5	−0.80 *	−0.21	−0.60	−0.36	<0.001
VERTIS § [44]	Ertuglifozin	Placebo	640/644	Yes	≥55	2.2	−0.33 *	0.08	−0.20	−0.06	<0.05
Other Interventions											
Bjorck [51]	Enalapril	Metoprolol	20/16	Yes	46 ± 14	3.0	−1.00 *	−0.01	−0.60	−0.093	0.021
RENAAL [52]	Losartan	Placebo	719/716	Yes	1.3–3.0	3.4	−0.77 *	−0.35	−0.53	−0.42	0.01
PIMA (ACR 30–299) [53]	Losartan	Placebo	39/39	Yes	167 ± 43	6.0	−5.20 *	−0.25	0.99	−0.45	0.42
MDRD study A [54]	LPD	Usual diet	287/291	No	25–55	3.0	−0.93 *	−0.21	−0.47	−0.32	0.30
	MAP < 92 mmHg	MAP < 107 mmHg	296/282	No	25–55	3.0	−0.90 *	−0.20	−0.50	−0.31	0.18
Nielsen [55]	Lisinopril	Atenolol	17/19	Yes	75 ± 6	3.5	−1.25	−0.59	−0.81	−0.54	0.63
SPS3 [56]	SBP < 130 mmHg	SBP 130–149 mmHg	1301/1309	Mixed	80 ± 19	5.0	−0.58	−0.10	−0.35	−0.07	<0.001
SPRINT-CKD [57]	SBP < 120 mmHg	SBP < 140 mmHg	1330/1316	No	20–59	3.3	−0.25 *	−0.04	0.28	−0.03	0.03

RCTs, randomized controlled trials. † Numbers refer to patients included in the GFR analysis; ^ Data are for patients randomized to Empaglifozin 25 mg/day. § Data are for patients randomized to ertuglifozin 15 mg/day. * Significantly greater difference versus control arm of initial eGFR dip. LPD, low-protein diet; MAP, mean arterial pressure, SBP, systolic blood pressure. ° Overall P refers to the comparison of GFR change between active and placebo arms throughout the whole period of follow up.

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
