# Peer review of "Nephroprotection by SGLT2 Inhibition: Back to the Future?"

_jcm, 2020, doi:10.3390/jcm9072243_

Round 1

Reviewer 1 Report

Dear Authors,

Presented paper is well-written, describes significant medical matter, and looks suitable for publishing in JCM.

Unfortunately, this manuscript shares too many common data and analyses with your previously published paper in the journal Medicine (Kaunas)

10.3390/medicina55060268

You almost duplicate the whole table 1 as soon as your paper is based on almost the same conceptualization. 

Nowadays, publishing in the scientific journals requires providing readers with fresh and sharp insights into exact medical field. In this terms, your manuscript does not provide significant input.

Taking into account the lack of novelty and ridiculous overlapping papers this review article have to be rejected at this time.

Author Response

Dear Authors,

Presented paper is well-written, describes significant medical matter, and looks suitable for publishing in JCM.

Unfortunately, this manuscript shares too many common data and analyses with your previously published paper in the journal Medicine (Kaunas).10.3390/medicina55060268.

You almost duplicate the whole table 1 as soon as your paper is based on almost the same conceptualization.

Nowadays, publishing in the scientific journals requires providing readers with fresh and sharp insights into exact medical field. In this terms, your manuscript does not provide significant input.

Taking into account the lack of novelty and ridiculous overlapping papers this review article have to be rejected at this time.

R: We thank the Referee for his comments.

Although the issue addressed by the two papers is the same (SGLT2i and Kidney), the current ms differs in a significant way from the previous one (that we now quote as last ref in the list). However, your critique led us to further modify the current paper. Therefore, we have now deleted the previous Table 1, added a new fully original Figure 2 (descriptive of check-mark), and provided more comments and insights into the core of this review, that is, correction of hyperfiltration as major mechanism of nephroprotective interventions, either novel (SGLT2i) or old (BP lowering drugs, anti-RAS, dietary salt restriction).  We believe that, also thank to your helpful comment, the final message on “back to the future” is now clearer.

Reviewer 2 Report

Summary:

This review summarizes the current knowledge on renal effects of SGLT2 inhibition. The main clinical studies on this aspect are presented, which uniformly demonstrate inhibition of progression in diabetic kidney disease. The potential underlying mechanisms are discussed with focus on attenuation of hyperfiltration, which is the predominant one. The review further compares the data from SGLT2 inhibitors to other forms of progression inhibition in CKD, such as RAS inhibition, which also lead to reduction in hyperfiltration, all accompanied by the “check mark sign”, the initial drop in eGFR. The manuscript thereby highlights the necessity of correcting hyperfiltration as the principal target of all current and future concepts of progression inhibition.

General comments:

This review is an interesting summary of the role of reduction of hyperfiltration as the key mechanism of progression inhibition by SGLT2 blockade. Even though the mechanism and its importance are well known – at least to nephrologists – it summarizes current considerations on this aspect and expands them by the idea of the “check mark sign” as an early indicator of successful progression inhibition. I think the review is worth reading. I have few suggestions for improvement, the major being that, in my opinion, it needs to be more concise in some passages. “Current unmet needs” (#2), for example, should be discussed more superordinate (not just RAS inhibition), whereas the current information could be included in the later part of the manuscript. #4, 6 and 7 could be shortened and less in detail, such that reading of the manuscript is facilitated to non-nephrologists, the principal readership of the journal.

Specific comments:

  • When dealing with potential barriers of SGLT2 inhibition, the most common side effects should be mentioned: urogenital infections and diabetic ketoacidosis (e.g. line 306). With all the positive results from the trials, the unexperienced physician needs to know about them for differentiated therapy. While beneficial for many patients, not every patient is eligible for SGLT2 inhibition.
  • Page 2, line 47, first sentence of the paragraph: Information about the investigated matter of the RCTs needs to be mentioned. There is some information lacking in this sentence.
  • Line 108 (page unclear), whole chapter #4: Most readers will not be used to the abbreviations. I suggest deleting most of them, to make it easier to understand these important considerations.
  • Line 164: The idea of different mechanisms in type 1 and type 2 diabetes is disturbing, especially, as you discuss the effect on afferent arterioles also in type 2 diabetes in earlier parts of the manuscript. Therefore, this finding needs to be explained
  • Line 234: I disagree on this statement, as reduction in proteinuria is also an effect of reduced net glomerular filtration pressure and thus reduced glomerular hyperfiltration.

Author Response

General comments:

This review is an interesting summary of the role of reduction of hyperfiltration as the key mechanism of progression inhibition by SGLT2 blockade. Even though the mechanism and its importance are well known – at least to nephrologists – it summarizes current considerations on this aspect and expands them by the idea of the “check mark sign” as an early indicator of successful progression inhibition. I think the review is worth reading. I have few suggestions for improvement, the major being that, in my opinion, it needs to be more concise in some passages. “Current unmet needs” (#2), for example, should be discussed more superordinate (not just RAS inhibition), whereas the current information could be included in the later part of the manuscript. #4, 6 and 7 could be shortened and less in detail, such that reading of the manuscript is facilitated to non-nephrologists, the principal readership of the journal.

R: We thank the Referee for the general comments that helped us better focusing on the central issue of the paper. Accordingly, we have (a) deleted the paragraph on “unmet needs” and included the discussion on efficacy and safety of RASi in the new paragraph #2; (b) shortened the new paragraphs 5 and 6 (previous #6 and 7), (c) added a new paragraph (new #7) to hopefully make the hypothesis more comprehensible to non-nephrologists. Please note that we could not shorten the new paragraph #3 (previous #4) because we had to take into account also the comments of Referee 3 that asked us to add more information.

Specific comments:

  • When dealing with potential barriers of SGLT2 inhibition, the most common side effects should be mentioned: urogenital infections and diabetic ketoacidosis (e.g. line 306). With all the positive results from the trials, the unexperienced physician needs to know about them for differentiated therapy. While beneficial for many patients, not every patient is eligible for SGLT2 inhibition.R: According to this request, we have now added this information in the new paragraph #2 (middle of first page of par.)
  •  
  • Page 2, line 47, first sentence of the paragraph: Information about the investigated matter of the RCTs needs to be mentioned. There is some information lacking in this sentence.R: This is a nice review by Chan GC and Tang SC (NDT 2016) that describes the unmet needs in nephrology practice due to two reasons: 1) the limited efficacy and/or low tolerability of traditional nephroprotective interventions, as emerged in as many as 23 landmark trials in diabetic patients with and without CKD, and 2) the absence of clinical benefits of new agents with positive experimental data. We have therefore expanded in this revised version the comment according to your request (first sentence of new paragraph #2).
  •  
  • Line 108 (page unclear), whole chapter #4: Most readers will not be used to the abbreviations. I suggest deleting most of them, to make it easier to understand these important considerations.R: We have modified the title of the previous paragraph 4 (now #3) and deleted “unusual” abbreviations throughout the text of this paragraph.
  •  
  • Line 164: The idea of different mechanisms in type 1 and type 2 diabetes is disturbing, especially, as you discuss the effect on afferent arterioles also in type 2 diabetes in earlier parts of the manuscript. Therefore, this finding needs to be explainedR: We fully agree with this comment and deleted the “disturbing” sentence also considering that SGLT2i are currently not indicated in type 1 diabetes
  •  
  • Line 234: I disagree on this statement, as reduction in proteinuria is also an effect of reduced net glomerular filtration pressure and thus reduced glomerular hyperfiltration.R: Thank you for this comment. We have now deleted the sentence in the new paragraph 5 and also modified figure 3 by evidencing the antialbuminuric effect of reduced hyperfiltration.
  •  

Reviewer 3 Report

The present review by De Nicola and coworkers comprehensively discusses  current evidence (both trial- and real world-data based) for SGLT2i use in the treatment of diabetic and non-diabetic CKD. Underlying nephroprotective mechanisms are discussed with a special focus on hemodynamic mechanisms. 

The review is adequately stuctured, well-written and up-to-date. I only have minor suggestions:

I generally agree with the authors that SGLT2i-mediated changes in glomerular hemodynamics and the important role of the tubulointerstitium in this regard are of utmost importance for the nephroprotection seen with this type of drugs.

However, the review only briefly discusses important direct effects of SGLT2i on inflammatory and pro-fibrotic processes.

In line 123 it is stated:  ......"potential suppression of mediators of inflammation and fibrosis and the reduction of oxidative and endoplasmic reticulum stress due to the increment in autophagic flux in podocytes and renal tubules [32]."

Citation 32, however, only covers the (highly interesting) aspect of reduced oxidative and ER stress due to changes in autophagic flux in podocytes and renal tubules, while evidence for the aforementioned anti-fibrotic or anti-inflammatory effect of SGLT2i is not cited.

Due to the robust SGLT2i-nephroprotection seen in diabetic and presumeable non-diabetic patients it is - based on the multifactorial etiology of CKD - likely that SGLT2i might have direct beneficial effects at the cellular level - even beyond altered hemodynamics - that could account at least for the long-term presevation of kidney function. For example, it has been recently demonstrated that SGLT2i block basal and TGF-β1-induced expression of key mediators of renal fibrosis and kidney disease progression, namely thrombospondin 1, tenascin C and PDGF-B, in two independent human proximal tubular cell lines (Am J Physiol Renal Physiol. 2019 Mar 1;316(3):F449-F462. doi: 10.1152/ajprenal.00431.2018: Unraveling Reno-Protective Effects of SGLT2 Inhibition in Human Proximal Tubular Cells.)

The authors might discuss and cite direct nephroprotective effects of SGLT2 at renal cell level in more detail.

Author Response

The present review by De Nicola and coworkers comprehensively discusses current evidence (both trial- and real world-data based) for SGLT2i use in the treatment of diabetic and non-diabetic CKD. Underlying nephroprotective mechanisms are discussed with a special focus on hemodynamic mechanisms.

The review is adequately stuctured, well-written and up-to-date. I only have minor suggestions:

I generally agree with the authors that SGLT2i-mediated changes in glomerular hemodynamics and the important role of the tubulointerstitium in this regard are of utmost importance for the nephroprotection seen with this type of drugs.

However, the review only briefly discusses important direct effects of SGLT2i on inflammatory and pro-fibrotic processes.

In line 123 it is stated: ......"potential suppression of mediators of inflammation and fibrosis and the reduction of oxidative and endoplasmic reticulum stress due to the increment in autophagic flux in podocytes and renal tubules [32]."

Citation 32, however, only covers the (highly interesting) aspect of reduced oxidative and ER stress due to changes in autophagic flux in podocytes and renal tubules, while evidence for the aforementioned anti-fibrotic or anti-inflammatory effect of SGLT2i is not cited.

Due to the robust SGLT2i-nephroprotection seen in diabetic and presumeable non-diabetic patients it is - based on the multifactorial etiology of CKD - likely that SGLT2i might have direct beneficial effects at the cellular level - even beyond altered hemodynamics - that could account at least for the long-term presevation of kidney function. For example, it has been recently demonstrated that SGLT2i block basal and TGF-β1-induced expression of key mediators of renal fibrosis and kidney disease progression, namely thrombospondin 1, tenascin C and PDGF-B, in two independent human proximal tubular cell lines (Am J Physiol Renal Physiol. 2019 Mar 1;316(3):F449-F462. doi: 10.1152/ajprenal.00431.2018: Unraveling Reno-Protective Effects of SGLT2 Inhibition in Human Proximal Tubular Cells.)

The authors might discuss and cite direct nephroprotective effects of SGLT2 at renal cell level in more detail.

R: We thank this Referee for the positive and constructive comments. Accordingly, we have now modified the new paragraph #3 (previous #4) and add a statement (with relative ref #32) on the SGLT2i effects at the cellular level. We could not expand more this paragraph because Referee 2 actually asked to shorten it.

Round 2

Reviewer 1 Report

X

Author Response

Referee 2

Dear authors,

thank you very much for your revised manuscript. The changes you have made have substantially improved the manuscript. I have two small and minor issues remaining:

line 274 ff: The study on acetazolamide is somewhat out of context for non-nephrologists. Please add some information on the mechanism of acetazolamide.

R: Thank you. You are completely right: we did not specify that acetazolimide is a prox tubule diuretic. We have now added thi information (line 163 of R2 version).

line 371/372: I appreciate your shortening of the manuscript. I would, however, include a small sentence/paragraph (without going too much into detail) about GLP1-analogues showing basically improvement of proteinuria, in comparison to SGLT2 inhibitors which also improve kidney function.

R: Thanks again. Your comment allowed us to better explain our thoughts that we originally had to delete because of paragraph shortening (please see lines 235-242 of R2 version).

Reviewer 2 Report

Dear authors,

thank you very much for your revised manuscript. The changes you have made have substantially improved the manuscript. I have two small and minor issues remaining:

  • line 274 ff: The study on acetazolamide is somewhat out of context for non-nephrologists. Please add some information on the mechanism of acetazolamide.
  • line 371/372: I appreciate your shortening of the manuscript. I would, however, include a small sentence/paragraph (without going too much into detail) about GLP1-analogues showing basically improvement of proteinuria, in comparison to SGLT2 inhibitors which also improve kidney function.

Author Response

(The authors gave the same response as above.)
